# On the Effects of Lateral Openings on Courtyard Ventilation and Pollution—A Large-Eddy Simulation Study

**Tobias Gronemeier** *,† and **Matthias Sühring** †

Institute of Meteorology and Climatology, Leibniz University Hannover, 30419 Hannover, Germany; suehring@muk.uni-hannover.de

* Correspondence: gronemeier@muk.uni-hannover.de; Tel.: +49-511-762-3232

† These authors contributed equally to this work.

**Abstract:** Courtyards are an omnipresent feature within the urban environment. Residents often use courtyards as recreation areas, which makes them crucial for the physical and psychological comfort of the urban population. However, considering that courtyards represent enclosed cavities, they are often poorly ventilated spaces and pollutants from neighboring traffic, once entrained, can pose a serious threat to human health. Here, we studied the effects of lateral openings on courtyard pollution and ventilation. Therefore, we performed a set of large-eddy simulations for idealized urban environments with different courtyard configurations. While pollutant concentration and ventilation are barely modified by lateral openings for wide courtyards, lateral openings have a significant effect on the mean concentration, the number of high-concentration events and the ventilation within narrower and deeper courtyards. The impacts of lateral openings on air quality within courtyards strongly depend on their orientation with respect to the flow direction, as well as on the upstream flow conditions and upstream building configuration. We show that lateral openings, in most cases, have a negative impact on air quality; nevertheless, we also present configurations where lateral openings positively impact the air quality within courtyards. These outcomes may certainly contribute to improve future urban planning in terms of health protection.

**Keywords:** courtyard; lagrangian particle model; large-eddy simulation; pollution; urban environment; ventilation

---

## 1. Introduction

Courtyards are an essential part of the urban environment. They serve as recreation areas for the local population and therefore play an important role for physical and psychological well-being in residential areas [1]. Courtyards are, however, often poorly ventilated spaces [2], where contaminants can pose a serious threat to human health once they are inside the courtyard cavity; with sources of contaminants could be external, such as from traffic on the nearby streets, or local, such as from domestic fuel [3] or from car parks within the courtyard. Beside the pollutant concentration also the time humans are exposed to the pollutants are critical factors that need to be considered in terms of human health protection [4,5]. For a street canyon, Lo and Ngan [6] showed that a significant amount of pollutants resides within the canyon for a longer time period (e.g., >10 min). Compared to street canyons, however, courtyards might be even worse ventilated since they have typically fewer exits. Hence, once pollutants are entrained into courtyards, they may reside within the courtyard for even longer compared to a street canyon. To design and improve urban planning in terms of health protection, it is, therefore, crucial to understand how contaminants are mixed into and out of the courtyards and for how long they reside within the courtyard cavities.

Although several studies have already described ventilation and pollutant removal from courtyards, e.g., [2,7–11], only a few address the influence of openings on courtyard ventilation. Courtyard ventilation itself largely depends on building configuration, e.g., aspect ratio and building height [2], wind speed [9], wind direction [10], stratification [2,8], as well as blocking obstacles within the courtyard itself [2]. Based on wind-tunnel experiments, Hall et al. [2] revealed that also tunnel-like lateral openings can significantly affect courtyard ventilation. They analyzed removal of pollutants released within the courtyards and showed that lateral openings can either increase or decrease the concentration compared to a closed courtyard, depending on the orientation of the opening with respect to the wind direction. They attributed this to the re-circulation within the courtyard cavity, which can be perturbed but also reinforced by cross flows induced by the opening, worsening or improving courtyard ventilation, respectively. However, in their experiments, Hall et al. [2] considered only undisturbed oncoming flow reaching the building block, which is largely different to a complex flow within an urban environment where neighboring buildings can significantly modify the flow field, e.g., [12,13]. Also, pollutants were only emitted within the courtyard cavity, leaving open the question of how much courtyard cavities are polluted from sources outside, e.g., from the adjacent street canyons?

Using large-eddy simulation (LES), Kurppa et al. [14] studied the effect of different city-block designs on pollutant dispersion. Although pollution within courtyards was only low compared to the adjacent street canyons, different concentrations were observed between the different building setups that partly included also gaps within the building blocks. By means of wind-tunnel experiments Ok et al. [7] showed that lateral openings can strongly affect the wind speed within the courtyards. The lowest wind speeds were observed within closed courtyards, while the highest were observed when multiple openings are aligned along the mean wind direction. Openings orientated perpendicularly to the main flow direction increased wind speeds less compared to other opening setups. Even though Ok et al. [7] did not investigate pollutant dispersion, their results indicate that lateral openings can significantly affect courtyard ventilation, which in turn might possibly also affect the air quality within courtyards.

The impact of pollutants on human health depends, among other aspects, largely on the concentration, e.g., [15]. Within busy street canyons, concentration levels can reach high values. In the absence of lateral building openings, significantly lower concentrations can be observed in the rear of the buildings [16] and adjacent backyards [8]. Before polluted air can reach these areas, it first needs to exit the street canyon via the roof level, where it is mixed with fresh air from above, leading to significantly lower pollutant concentrations. However, this might become different when lateral openings are present, where high pollutant concentrations can be directly mixed into the courtyards.

Here, motivated by the findings of previous studies, we ask:

- What is the effect of lateral openings on courtyard pollution and ventilation within an urban environment?
- How do lateral openings affect maximum concentrations and residence time scales within courtyards?

To answer these questions, we used LES datasets for idealized building arrays to investigate pollutant dispersion into courtyards. Pollutant sources are considered outside of the courtyard cavity resembling e.g., car exhausts from streets. We considered different building configurations, i.e., different aspect ratios, and different orientation of lateral tunnel-like openings with respect to the mean wind. Although observations already revealed that stratification also significantly influences pollutant dispersion in urban environments [8], we exclusively concentrate on neutral conditions within this first study.

Section 2 describes the LES model, the simulation setups as well as the applied analysis techniques followed by validation results of the LES model against wind-tunnel data. Section 3 gives a description of the mean flow field and concentration distribution and shows results on the net transport of scalar

through the openings as well as the analysis of high-concentration events and residence times. Finally, Section 4 gives a summary and closes with ideas for future studies.

## 2. Methods

### 2.1. LES Model and Numerical Experiments

For the numerical simulations in our study, we used the LES model PALM [17], revision 2705. PALM has been already successfully used to simulate the flow in urban environments in high detail, e.g., [14,18–22]. Also, PALM provides the possibility to represent three-dimensional building topologies and includes an embedded Lagrangian particle model, making it well-suited to study pollutant dispersion and ventilation in urban environments. PALM solves the non-hydrostatic incompressible Boussinesq equations. For the subgrid model, the kinetic energy scheme of Deardorff [23] was used. The advection terms were discretized by a fifth-order scheme [24], while near solid walls the order of the scheme was successively degraded. For the time discretization a third-order Runge-Kutta scheme by Williamson [25] was used.

The model domain consists of several building patches that are aligned and shifted in rows as depicted in Figure 1. A single patch consists of a building containing a courtyard and an adjacent street at its southern and western wall. To study effects of different building/courtyard geometries on courtyard ventilation, we performed three simulations with different courtyard aspect ratio (AR, ratio of building height, or courtyard depth, $H$, to courtyard width $W$), which are listed in Table 1. The case with AR = 1, where $H = W$, as well as the cases with high and low AR are hereafter referred to as "AR1", "AR3" and "AR03", respectively. The case AR1 was chosen to link to other studies as this is the most famous case throughout other research, e.g., [2,7,9,10]. The other two cases, AR03 and AR3, were chosen as they showed the most pronounced differences in scalar concentrations compared to the AR1 case within the study by Hall et al. [2]. The building within a patch has a single lateral opening either on its western, eastern, northern or southern wall, or has no opening at all, i.e., it is closed. These patches are labelled as "W", "E", "N", "S", and "C", respectively. The lateral opening has a size of 4 m by 4 m and is always located at the bottom center of the respective building wall to represent an entrance to the courtyard. The size of the opening is chosen according to our personal experience, assumed to be typical for mid-European city quarters, even though we note that openings vary in size in real cities, depending on the prevailing architecture. Also, different sized openings at other locations of a building wall might also appear within a real city. However, in this idealized study, we tried to keep the setup simple, to limit the number of effects and hence the complexity of the results.

**Table 1.** Courtyard aspect ratio (AR) and domain size of the three simulated cases. $H$ indicates the building height (or courtyard depth) and $W$ indicates the courtyard width.

| Case | AR | $H$ (m) | $W$ (m) | Domain Size ($x \times y \times z$) (m) |
|------|-----|---------|---------|------------------------------------------|
| AR1 | 1 | 20 | 20 | $480 \times 400 \times 531$ |
| AR3 | 3 | 60 | 20 | $480 \times 400 \times 531$ |
| AR03 | 0.3 | 20 | 60 | $480 \times 600 \times 531$ |

A row of buildings is then formed by aligning five building patches, with each patch along the $y$-direction having a different courtyard/opening configuration (see Figure 1). The street width was set to 20 m for all streets and the building-wall thickness was set to 20 m for all buildings in all simulated cases.

For cases AR1 and AR3, the domain consists of six rows, while three neighboring rows are shifted along the $y$-direction, forming a front, center and back row with respect to the $x$-parallel flow from the West (see Figure 1b). For case AR03, the center rows are missing resulting in only four rows within the domain (see Figure 1c). This was done to keep the length of the $x$-parallel (wind-parallel) streets constant throughout the different cases (compare Figure 1b,c). This ensures that in all simulated setups

the flow can accelerate the same distance and has the same input of scalar (see details below) along the *x*-parallel streets.

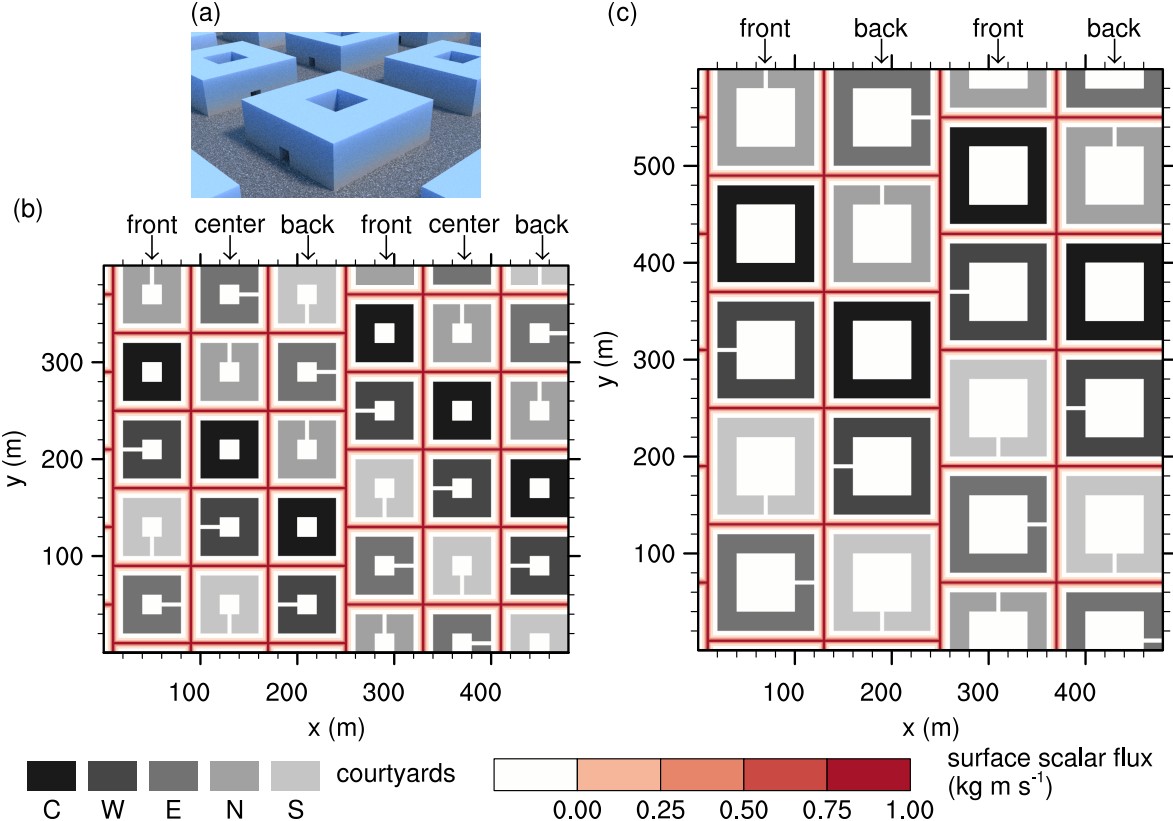

**Figure 1.** (**a**) 3D view of the building setup used in case AR1 and (**b**,**c**) horizontal cross-sections of the simulation domains and building configuration: (**b**) case AR1 and AR3, (**c**) case AR03. Different shades of *grey* indicate building patches with a different courtyard configuration, labelled according to their lateral opening orientation. *Red* colors indicate locations and strength of the scalar sources. The "front", "center" and "back" labelling indicates the front-, center-, and back-building row of the staggered building patches.

We have chosen this staggered building setup to break-up the street canyons along the *x*-direction, to prevent artificial jets that would develop along the infinite *x*-parallel street canyons, as we did observe in preparatory test simulations. Hence, we optimized our setup to prevent such unrealistic long streets. Furthermore, these setups allow study of different courtyard realizations within a single simulation, which requires significantly less computational effort compared to study each courtyard realization in a single simulation individually, which would not be possible without increasing the grid spacing. To study different courtyard realizations in one simulation, however, requires that the flow and scalar distribution within the different courtyard cavities are statistically independent from each other. Indeed, an analysis of velocity and scalar variances within identical courtyards but different upstream courtyards revealed no significant differences (not shown). Hence, we are confident that the impact of a lateral opening is limited to the courtyard cavity itself and the part of the street directly adjacent to the opening. This is only true as long as lateral openings do not directly face each other, hence, we avoided this configuration in our building layout (see Figure 1).

At this point, we would like to note that the studied building configuration is highly idealized. In reality, buildings would vary in height and orientation, and courtyard configurations and opening sizes would be more heterogeneous. However, as we try to focus purely on effects of openings on courtyard ventilation and pollution, we idealized the building setup to isolate those effects, while still trying to mimic an urban environment with neighboring buildings.

In total, the domain size adds up to 480 m in the $x$-direction and 400 m (case AR03: 600 m) in the $y$-direction with a domain height of 531 m. A rectilinear grid with an isotropic grid spacing of 0.4 m within the lower 200 m of the domain was used. To save computational costs, the vertical grid spacing (along $z$-direction) above 200 m was stretched by a factor of 1.08 until it reached 4 m at a height of 247 m, from where on it was kept constant up to the domain top. Overall, the domain consisted of 1200 by 1000 by 602 grid cells (case AR03: 1200 by 1500 by 602) in $x$- $y$- and $z$-direction, respectively.

The simulations were initialized with a logarithmic wind profile reported by Hall et al. [2] and driven by a horizontal pressure gradient of $-1 \times 10^{-4}\,\mathrm{Pa\,m^{-1}}$ along the $x$-direction, resulting in a mean wind speed of $(1.9 \pm 0.3)\,\mathrm{m\,s^{-1}}$ at $z = 2H$ during the analysis period with flow parallel to the $x$-direction, while almost constant wind speed during the analysis period.

At the lateral boundaries, we used cyclic conditions at the spanwise boundaries and shifted cyclic conditions according to Munters et al. [26] at the streamwise boundaries. The shifted cyclic condition was used to prevent the generation of streamwise-elongated coherent structures that can appear if pure cyclic conditions are applied [26]. The shifting distance along the $y$-direction was set to the size of a single building patch, hence, to 80 m in case AR1 and AR3 and 120 m in case AR03. Free-slip boundary conditions were applied at the domain top. As surface boundary condition for the momentum equations (at Earth and building surfaces), Monin–Obukhov similarity theory (MOST) was applied locally between the surface and the first grid point normal to the respective surface orientation. This applies for all surfaces, i.e., at horizontal upward- and downward-facing surfaces (at the top of the lateral opening), as well as at vertical surfaces, following Park et al. [20] and Park et al. [27]. The boundary layer in our simulations is purely shear-driven, i.e., we solved no equation for the temperature or humidity.

To investigate dispersion of pollutants, e.g., from car exhausts, into the courtyards, we considered line sources of passive scalar within the street canyons, as indicated by the *red* lines in Figure 1. These sources emulate a time-constant, Gaussian-shaped surface scalar flux along the center line of the streets. This way, we simulated the pollutant release from traffic within the street canyons which allows us to investigate how such pollutants are transported into the courtyards. Surface fluxes were identical on all streets, i.e., we did not distinguish between main and side streets with different traffic density. Besides directly simulating pollutant sources, also other concepts exist which evaluate the ventilation of the urban environment such as the concept of air delay [28]. The air delay gives an estimate of how long a specific air parcel resides within the urban environment and this way concludes the ventilation. However, the main focus of this study is to evaluate how pollutants are advected from outer sources into the courtyard cavity while lateral openings are considered. Therefore, a direct simulation of scalar sources is superior to indirect measurements.

The total simulation time for all cases was 4 h. This includes 2 h spin-up time and 2 h analysis time.

Presented data were averaged over the analysis period (2 h) as well as over identical courtyards (in a simulation, there are two identical courtyard realizations). Before time-averaging, scalar concentration $s$ was normalized by the time-dependent background concentration $s_B$, which is defined as the domain-averaged concentration at $z = H$. The normalization was done to account for any time dependencies in scalar concentration due to the time-constant scalar flux.

To investigate the relative occurrence of high concentrations within the courtyards and how these depend on the lateral openings, we calculated probability density functions (PDF) for the scalar concentration. Concentrations were sampled at the courtyard center at a height of 1.8 m at each time step during the analysis period. The sampled concentrations were then normalized by $s_B$.

Finally, in the following, we refer to courtyards with westward lateral opening in the front, center, and back row as "W front", "W center", and "W back", respectively (equivalent for the other opening orientations, see Figure 1).

The used model parameter lists for PALM for all described cases, as well as the additional code parts used for data analysis in this study are included in the Supplementary Materials.

## 2.2. Balance Term Analysis

To study courtyard pollution in more detail and distinguish between mixing of scalar into the courtyard through the top opening and the lateral opening, we examined the terms of the time-averaged scalar transport equation

$$\frac{\overline{\partial s}}{\partial t} = -\frac{\partial \overline{u_{1,2}\,s}}{\partial x_{1,2}} - \frac{\partial \overline{u_3\,s}}{\partial x_3} - \frac{\partial \overline{\tau_{1,2,s}}}{\partial x_{1,2}} - \frac{\partial \overline{\tau_{3,s}}}{\partial x_3}, \tag{1}$$

where the left-hand side describes the local temporal change of passive scalar *s*. The first and the second terms on the right-hand side describe the resolved-scale transport of *s* in horizontal as well as in vertical direction, respectively, with $u_{1,2}$ being the horizontal velocity components and $u_3$ being the vertical component. The third and fourth terms on the right-hand side are the parametrized turbulent transport on the subgrid scale in horizontal ($\tau_{1,2,s}$) and vertical direction ($\tau_{1,s}$), respectively. The overbar indicates the time-averaging. To be consistent with the numerical discretization, we directly used the flux divergence provided by the advection scheme and the subgrid-scale parametrization. As no sources or sinks of passive scalar exist within the courtyard volume, the entire passive scalar is entrained into the courtyard via the openings, so that we can make use of Gauss's theorem to calculate the net transport. Thus, integrating Equation (1) over the entire courtyard volume leads to the mean net transport of scalar along the respective spatial direction,

$$\int_V \frac{\overline{\partial s}}{\partial t}\,\partial V = \int_V \left(-\frac{\partial \overline{u_{1,2}\,s}}{\partial x_{1,2}} - \frac{\partial \overline{\tau_{1,2,s}}}{\partial x_{1,2}}\right)\partial V + \int_V \left(-\frac{\partial \overline{u_3\,s}}{\partial x_3} - \frac{\partial \overline{\tau_{3,s}}}{\partial x_3}\right)\partial V, \tag{2}$$

with *V* indicating the entire courtyard volume up to $z = H$. The left-hand side describes the temporal mean accumulation of scalar within the courtyard volume, which is, however, relatively small compared to the terms on the right-hand side. The first term on the right-hand side of Equation (2) gives the net transport of scalar via the lateral opening ($T_l$), while the second term gives the vertical net transport of scalar via the top opening ($T_v$). A positive value indicates an increase of passive scalar, while a negative value indicates a decrease of passive scalar. For closed courtyards, the $T_l$ vanishes so that scalar accumulation is only due to the vertical transport.

## 2.3. Evaluation of Pollutant Residence Times

The impact of pollutants on human health depends, among other factors, on the time humans are exposed to these pollutants [4,5], or, in other words, how long pollutants reside within the courtyard cavity. To estimate the residence times of pollutants within courtyards, we followed Lo and Ngan [6] and used a Lagrangian particle model embedded into the LES model, where the residence time is defined as the time elapsed between the entry of a particle into the region of interest and its exit [29]. Although the Lagrangian particle model requires higher computational resources, it allows us to directly measure the residence time and therefore gives more reliable results than indirect measurements retrieved from scalar concentration values as, e.g., when analyzing the air delay.

The embedded Lagrangian particle model is based on Weil et al. [30], to separate the particle speed into a deterministic and a stochastic contribution, which corresponds to dividing the turbulent flow field into a resolved-scale and a subgrid-scale (SGS) portion, respectively, following the LES philosophy. The resolved-scale velocity is provided by the LES at each time step, while the SGS velocity is predicted by integrating a stochastic differential equation according to Weil et al. [30], who strictly adopted the Thomson [31] model to the subgrid scale by assuming isotropic and Gaussian-distributed turbulence. To parametrize the stochastic particle dispersion on the subgrid scale, the LES provides local values of the SGS turbulent kinetic energy and the dissipation rate at each time step.

According to Steinfeld et al. [32], the LES data are interpolated bi-linearly on the actual particle position in the horizontal. In the vertical, a linear interpolation is used, except for the particles located between the surface and the first grid level, where a logarithmic interpolation according to local MOST

for the resolved-scale horizontal velocity components is applied. At the solid boundaries, i.e., upward- and downward-facing as well as vertical building surfaces, we used a reflection boundary condition for the particles, and cyclic conditions at the lateral boundaries. A more detailed description of the particle model embedded into the LES model is given by Steinfeld et al. [32] and Maronga et al. [17].

Following Lo and Ngan [6], we calculated the residence time of a particle by summing-up the total time the particle spent within the courtyard volume. Once a particle exits the courtyard volume by the lateral or the top opening, the particle age is stored and the particle itself is immediately removed from the simulation. Particles were released within the courtyard each LES time step at a height of 1.8 m. This particle-source height should be representative for human exposure. Setting the particle sources within the courtyard only, has the advantage that fewer particles need to be modelled to obtain sufficient statistics, compared to the case where particle sources are along the street canyons and only a small portion of the particles would be mixed into the courtyards.

Physically, larger residence times indicate less turbulent mixing and ventilation of the courtyard. This in turn indicates larger impact of pollutants on human health in case of high concentrations, compared to smaller residence times [6]. Please note that we focus on residence times only, which is a pure ventilation measure. Following Lo and Ngan [6], however, the exposure time is a more direct measure to relate the impact of pollutants on human health as it also considers re-entrainment of particles into the region of interest, which the residence time does not. However, Lo and Ngan [6] showed that the number of re-entrainment events is relatively small, so that we decided to focus on the residence times, for the sake of simplicity and computational effort.

At this point, we want to note that Lo and Ngan [6] did observe particle accumulation near solid walls in their study (also using PALM) and excluded these regions from their analysis. In preparatory studies, we could observe similar particle accumulation near solid walls. This accumulation could be traced back to an erroneous treatment of SGS particle velocities near solid walls, which was fixed in PALM revision 2418. In the following, we could not observe particle accumulation near solid walls any more.

### 2.4. Validation and Grid Sensitivity

To prove PALM's capability to correctly represent the flow within a courtyard cavity, we first compare simulation results against data from wind-tunnel experiments by Hall et al. [2] and Reynolds-averaged simulations by Ryu and Baik [9]. The validation setup consists of a single building with a closed courtyard (no lateral opening) and an undisturbed incoming flow according to the wind-tunnel experiments shown by Hall et al. [2]. The building setup (height, width) is similar to a single building of case AR1 with its center placed at $(x, y) = (370 \, \text{m}, 144 \, \text{m})$. The domain size for this case is 620 m by 288 m by 240 m in the $x$-, $y$-, and $z$-direction, respectively, with an isotropic grid spacing of 0.4 m. The simulation time was 4 h. The complete parameter list of this simulation is included in the Supplementary Materials.

Figure 2 shows profiles of the normalized and time-averaged $u$-component of the wind speed and its standard deviation. Within the courtyard cavity, $u$ agrees well with the data from Ryu and Baik [9] and Hall et al. [2], except for the lower half of the courtyard where Hall et al. [2] reported a higher negative $u$-component, indicating a stronger re-circulation within the courtyard cavity. Above the cavity, the simulated $u$-profile agrees fairly well with the observed wind-tunnel data. The variation $u'$ can only be compared with the wind-tunnel measurement of Hall et al. [2] as it was not reported by Ryu and Baik [9]. The simulated $u'/u_0$ is about 0.1 within the courtyard cavity, while Hall et al. [2] reported larger values of around 0.2 within the cavity. However, other courtyard setups with larger and smaller ARs reported by Hall et al. [2] showed significantly lower values which are in better agreement with our validation case. Since the mean wind profile is in good agreement with the data reported by Hall et al. [2] and Ryu and Baik [9] and $u'$ shows qualitatively good agreement with profiles of Hall et al. [2], we are confident that PALM is capable of correctly simulating the flow inside the courtyard cavity.

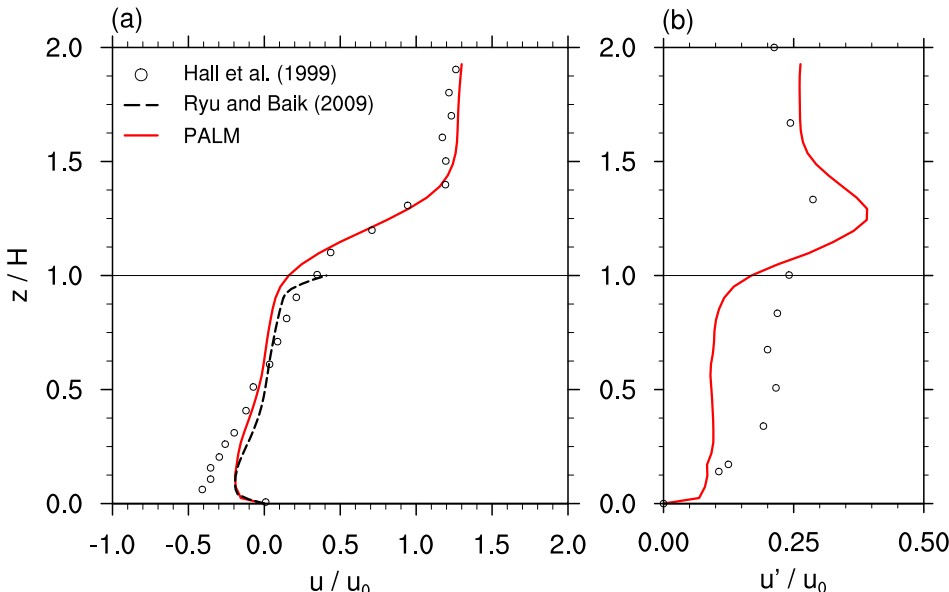

**Figure 2.** Vertical profiles of normalized (**a**) $u$-component of the wind and (**b**) its standard deviation $u'$ at the center of the courtyard. The red curve shows the profiles simulated by PALM, the black curve the simulated data by Ryu and Baik [9], and the dots data from the wind-tunnel experiments by Hall et al. [2]. $u_0$ represents the mean oncoming wind speed at $z = H$ (height marked by horizontal line). The PALM profiles are time-averaged over 3 h.

The scalar dispersion simulated by PALM was previously validated by Park et al. [20] via wind-tunnel data for a street-canyon case and is therefore not validated again in the current study.

By definition, the results of an LES with implicit filtering, as used in PALM, depend on the grid spacing [33,34]. In practice, the question is whether the analyzed statistical moments converge towards finer grid spacing. Hence, we performed a grid sensitivity study where we conducted a simulation of a single building patch as described above (domain size: 80 m × 80 m), including a courtyard with an opening in wind direction. Three different grid sizes (1 m, 0.4 m and 0.2 m) were considered and the simulation time was 10 h. Other simulation parameters were identical to the main simulations (a detailed parameter list is included in the Supplementary Materials).

Figure 3a,c show the mean wind speed within the courtyard cavity and along the center line of the opening, respectively. The mean wind speed differs most between the simulation with 1 m and 0.4 m grid spacing, while the differences between 0.4 m and 0.2 m grid spacing are only small. A similar behavior can be observed for the standard deviation $u'$ (Figure 3b,d), even though there are still small differences between 0.4 m and 0.2 m grid spacing. Hence, as a grid spacing of 0.2 m does not yield to significantly different results compared to a grid spacing of 0.4 m but would significantly increase the computational demands, we chose a grid spacing of 0.4 m for all following simulations.

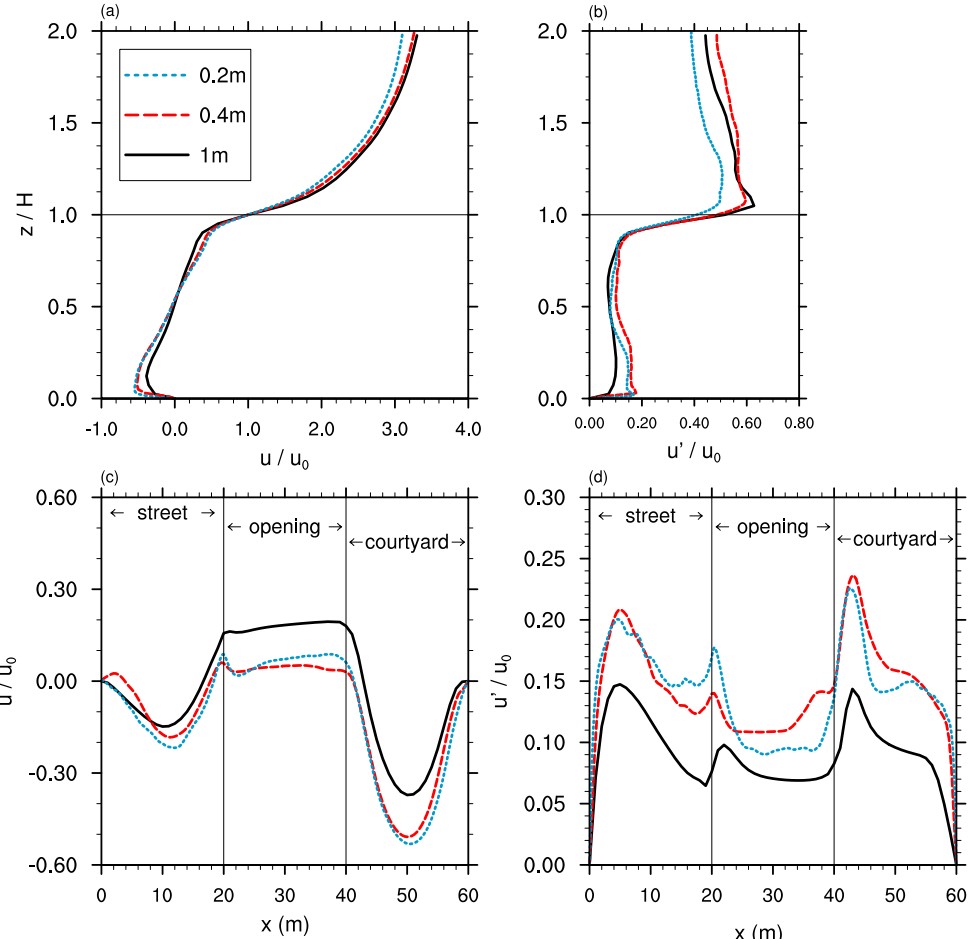

**Figure 3.** Vertical profiles of normalized (**a**) *u*-component of the wind and (**b**) its standard deviation $u'$ at the center of the courtyard, as well as horizontal profiles of (**c**) $u$ and (**d**) $u'$ along the center line of the courtyard opening for different grid sizes. $u_0$ represents the mean oncoming wind speed at $z = H$ (height marked by horizontal line). Profiles are time-averaged over 3 h.

## 3. Results

### 3.1. Mean Flow and Scalar Distribution

#### 3.1.1. Case AR1

Figures 4a–c and 5a–c show vertical and horizontal cross-sections, respectively, of the time-averaged flow field and scalar concentration for the closed courtyards in case AR1 (ref. Figure 1 and Table 1 for better orientation). Within the courtyard cavities, well-defined re-circulations are present which extend throughout the entire cavity (Figure 4a–c). Similar well-defined re-circulations can be observed within the street canyons between the buildings (Figure 4b) as well as downstream of the back-building row (Figure 4c), where the strongest re-circulations with the largest horizontal extent into downstream direction can be observed. Furthermore, upstream of "C front" we can observe a small eddy close to the surface, attributed to the non-blocked oncoming flow through the *x*-parallel street canyon (Figure 4a). All these mentioned flow patterns were already focused on in previous studies and are well documented, e.g., [12,13,35,36]. Furthermore, we refer also to the work of Hall et al. [2] and Ryu and Baik [9], where the flow pattern within closed-courtyard cavities is already well described.

The highest scalar concentrations can be observed within the street canyons (Figure 4). The largest values occur near the surface and the leeward walls of the canyon, according to the observed pattern in previous studies [12,37], as it becomes most obvious downstream of "C back" (Figure 4c), where the re-circulation can catch up more scalar along the *x*-parallel street canyon compared to the re-circulations

downstream of "C front" or "C center" (compare Figures 4a,b and 6a). Within the closed-courtyard cavities, the concentrations are significantly lower compared to those within the street canyons, with values slightly lower than $s_B$, while no significant differences can be observed among the front-, center-, and back-row courtyards.

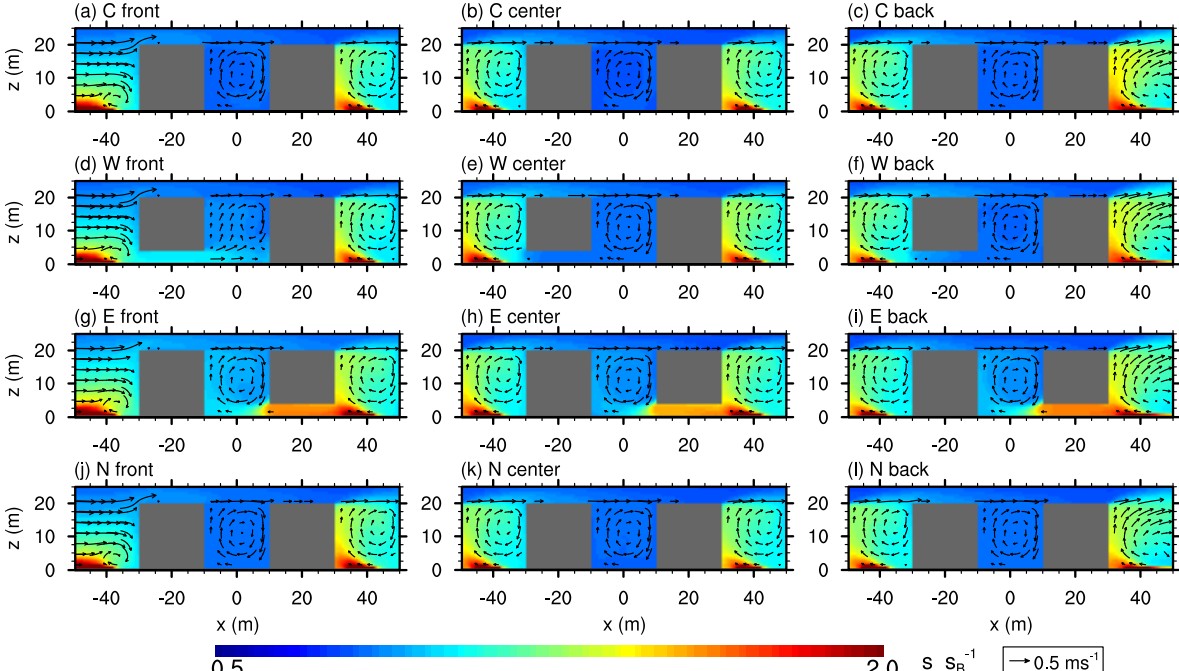

**Figure 4.** *Xz*-cross-section of the mean flow field (vector arrows) and mean scalar concentration (contours) at the courtyard center for case AR1. Scalar concentration is normalized with the background concentration $s_B$ at $z = H$. Please note, due to symmetry, courtyards with southern openings show similar scalar distribution and wind field than those with northern openings and are hence not shown.

If lateral openings are present, the flow and concentration patterns within the courtyards change, depending on the flow pattern within the adjacent relevant street canyon. For courtyard "W front", the opening faces the incoming flow from the upstream *x*-parallel street canyon. It can thus directly enter the courtyard, which leads to a significant input of scalar (see Figure 5d) with highest concentrations near the surface and the lateral opening. Within the "W front" courtyard, the re-circulation is still present but shifted upwards as well as towards the windward building wall (see Figure 4a,d) compared to that in the closed case, while in the lower part of the courtyard also a horizontal re-circulation forms (see Figure 5d). This perturbation of the re-circulation agrees with the findings of Hall et al. [2], who identified a perturbation of the re-circulation within the courtyard due to lateral openings.

For courtyards "W center" and "W back", similar re-circulation patterns can be observed within the street canyon and the courtyard cavity, with low concentrations inside the courtyards. This is attributed to the re-circulation within the street canyon (courtyard) which points away (towards) the opening (see Figure 4e,f) and so prevents entrainment of scalar into the courtyard cavity through the lateral opening.

In contrast, courtyards with an opening on their eastward side ("E front", "E center", and "E back") show a significantly higher scalar concentration (Figure 5g–i). In these cases, the circulation within the street canyons transports the polluted air towards the courtyard cavity (cf. Figure 4g–i), which might be further supported by the circulation inside the courtyard that further distributes the entrained scalar throughout the cavity.

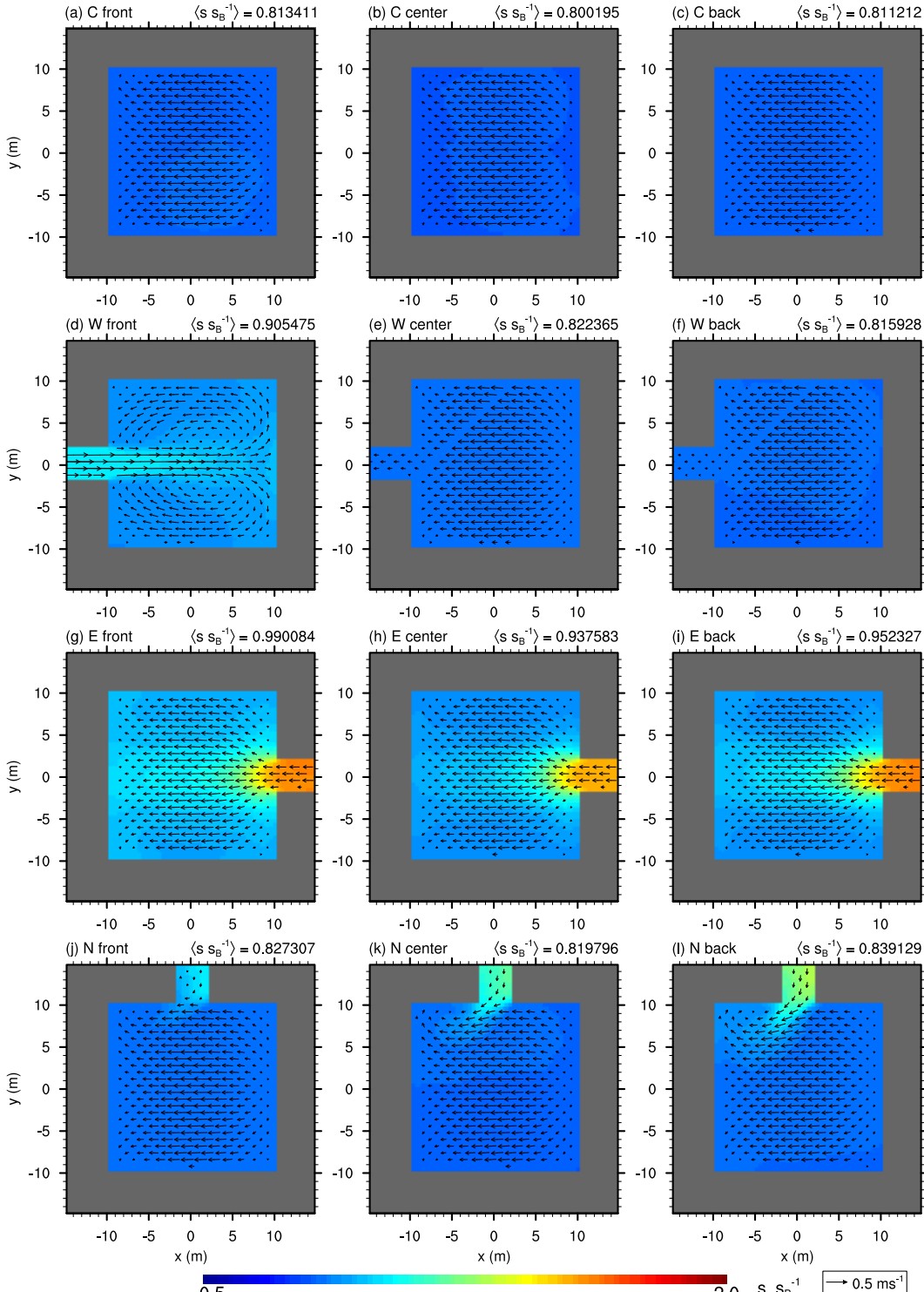

**Figure 5.** *Xy*-cross-section of the mean flow field (vector arrows) and scalar concentration (contours) within courtyards at $z = 1.8\,\mathrm{m}$ for case AR1. Scalar concentration is normalized with the background concentration $s_\mathrm{B}$ at $z = H$. Please note, due to symmetry, courtyards with southern openings show similar scalar distribution and wind field than those with northern openings and are hence not shown.

If the opening is located at the northern wall, the *x*-parallel street-canyon flow slightly pushes polluted air into the courtyard (cf. Figure 5j–l). The strength of this mean inflow as well as the scalar concentration within and near the lateral opening increases from the front row to the back row. However, it does not significantly affect the mean flow pattern within the rest of the courtyard and

the re-circulation is quite similar to that of a closed courtyard (cf. Figure 4j–l). The concentration is about the same or slightly higher within these courtyards compared to the closed ones as well as "W center" and "W back", but it is well below the levels of courtyard "W front" and those with openings at the eastward wall. Please note, due to symmetry, results for courtyards with openings located at the northern wall are similar to those with openings at the southern wall.

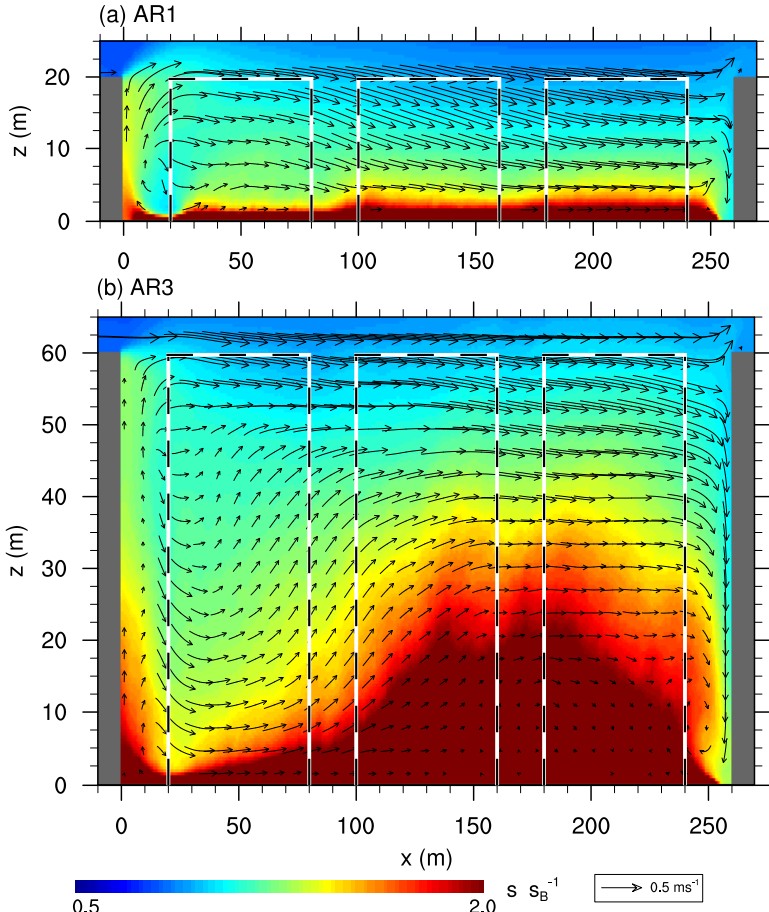

**Figure 6.** *Xz*-cross-section of the mean flow field (vector arrows) and scalar concentration (contours) along the center of an *x*-parallel street for case (**a**) AR1 and (**b**) AR3. Scalar concentration is normalized with the background concentration $s_B$ at $z = H$. The black-and-white lines indicate the positions of the buildings along the street.

### 3.1.2. Case AR3

Figure 7 shows vertical cross-sections for the high aspect ratio case AR3, where courtyards as well as the street canyons are three times deeper compared to AR1 (ref. Table 1). The re-circulation within the courtyard cavities does not extent throughout the entire cavity but occurs only within the upper part. Close to the surface, a weak secondary counter-rotating circulation is present, similar to the findings described in, e.g., Hall et al. [2] and Assimakopoulos et al. [12] (see Figure 7a–c). Similar to case AR1, the non-blocked oncoming flow along the *x*-parallel street canyon upstream of the front row causes a small eddy close to the surface at the windward building wall, where also air from the upper part of the street canyon is transported downwards (see Figure 7a). Likewise, downstream of the back row, a re-circulation downstream of the building can be observed. The street-canyon flow between the center and back row shows a comparable pattern as within the courtyard cavity. Between the front- and center-row building (see Figure 7b), it strikes that the street-canyon flow exhibits a vertical component which extends down to the surface, which might be linked to the flow along the *x*-parallel street canyon.

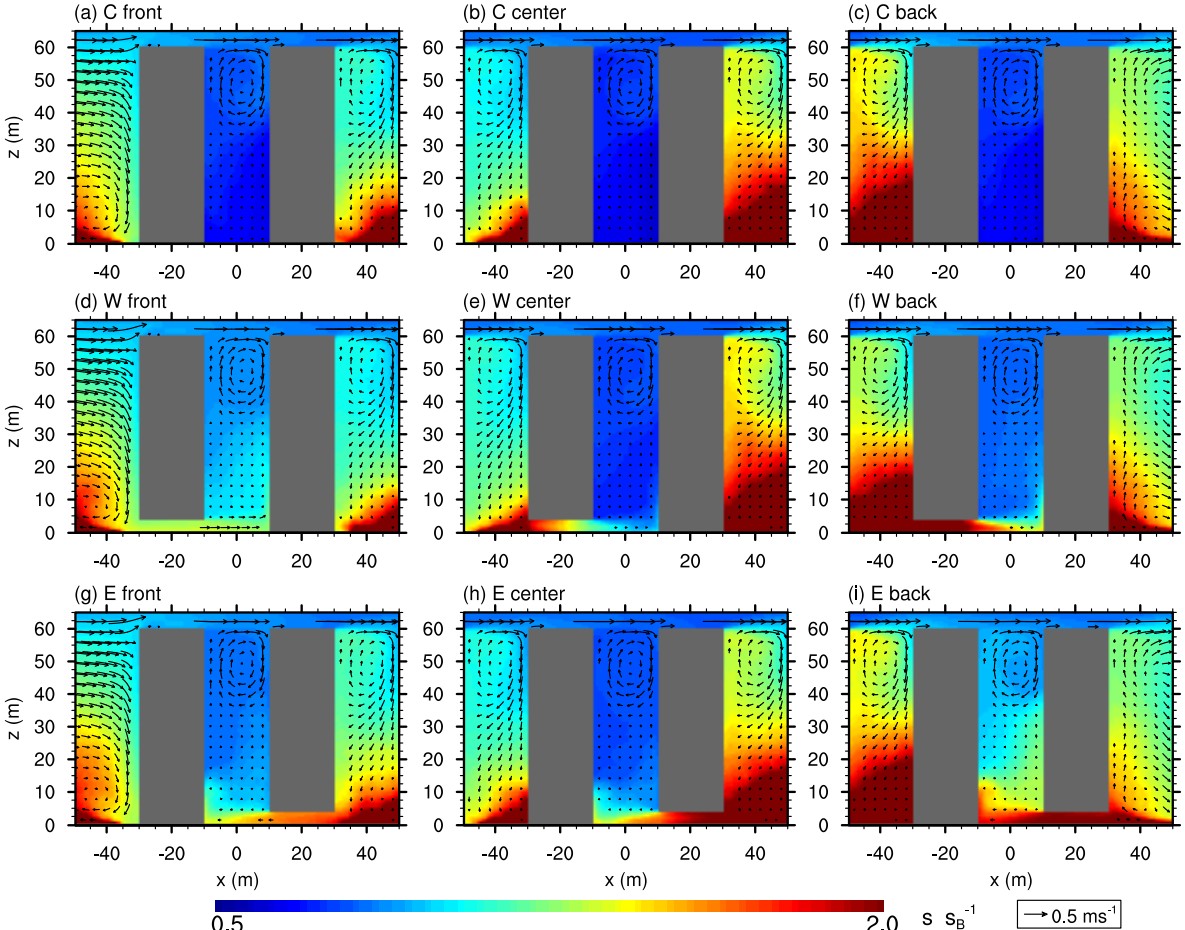

**Figure 7.** *Xz*-cross-section of the mean flow field (vector arrows) and mean scalar concentration (contours) at the courtyard center for case AR3. Please note, for reasons of space, not all realizations are shown.

The scalar concentration within the closed-courtyard cavities is lowest near the surface with values significantly lower than the background concentration; and highest in the upper part where scalar is vertically mixed into the cavity but not effectively transported downwards. Within the street canyons, the concentration shows significantly higher values compared to the courtyard cavity, while the scalar distribution is quite different for the different canyons. Upstream of the front-row building, the largest values can be observed in the lower part of the canyon further upstream of the street, while directly at the windward building wall, relatively lower concentrations can be observed which correlates with the downward transport of less-polluted air from above near the wall (see Figure 7a,d,g).

Between the front and the center row (Figure 7b) the highest concentrations occur near the surface at the windward building wall, which might be due to the downward component which is strongest near the leeward building wall where it transports fresh air downward (this, however, needs further investigation as it is beyond the scope of this study). Between the center- and back-row, scalar concentration is highest compared to the other street canyons along *y*-direction (see, e.g., Figure 7b,c), which is related to the flow and scalar concentration along the *x*-parallel street as depicted in Figure 6b. Similar to case AR1, the highest concentrations behind the back-row building can be observed at the leeward wall, attributed to the re-circulation (compare, e.g., Figures 4c and 7c).

The differences in the flow field and scalar concentration between the *y*-parallel streets, as indicated by Figure 7, might be also linked to the flow and scalar distribution along the *x*-parallel streets, which is depicted in Figure 6b. Compared to case AR1, the extent of the re-circulation downstream of the back-row building is significantly larger in case AR3, reaching approximately up

to the second $y$-parallel side street. This also affects the scalar distribution along the $x$-parallel street canyon, which is more heterogeneous compared to case AR1 (see Figure 6a).

All opened courtyards show an inflow through their lateral opening and hence a higher scalar concentration compared to the closed courtyards, as indicated by Figures 7 and 8. In contrast to the closed courtyards, the highest concentrations can be observed within the lower poorly ventilated parts of the cavities, whereas the better-ventilated upper parts of the cavities exhibit lower concentrations. For "W front", the oncoming flow along the $x$-parallel street enters the courtyard causing a significant increase of scalar concentration near the courtyard surface (see Figure 8a), similar to case AR1. Caused by the weak street-canyon circulation, "W center" and "W back" show a mean inflow into the courtyard cavity, with lower scalar concentration in case of "W center" attributed to the downward-mixing of less-polluted air from upper parts of the street canyon (Figure 7e).

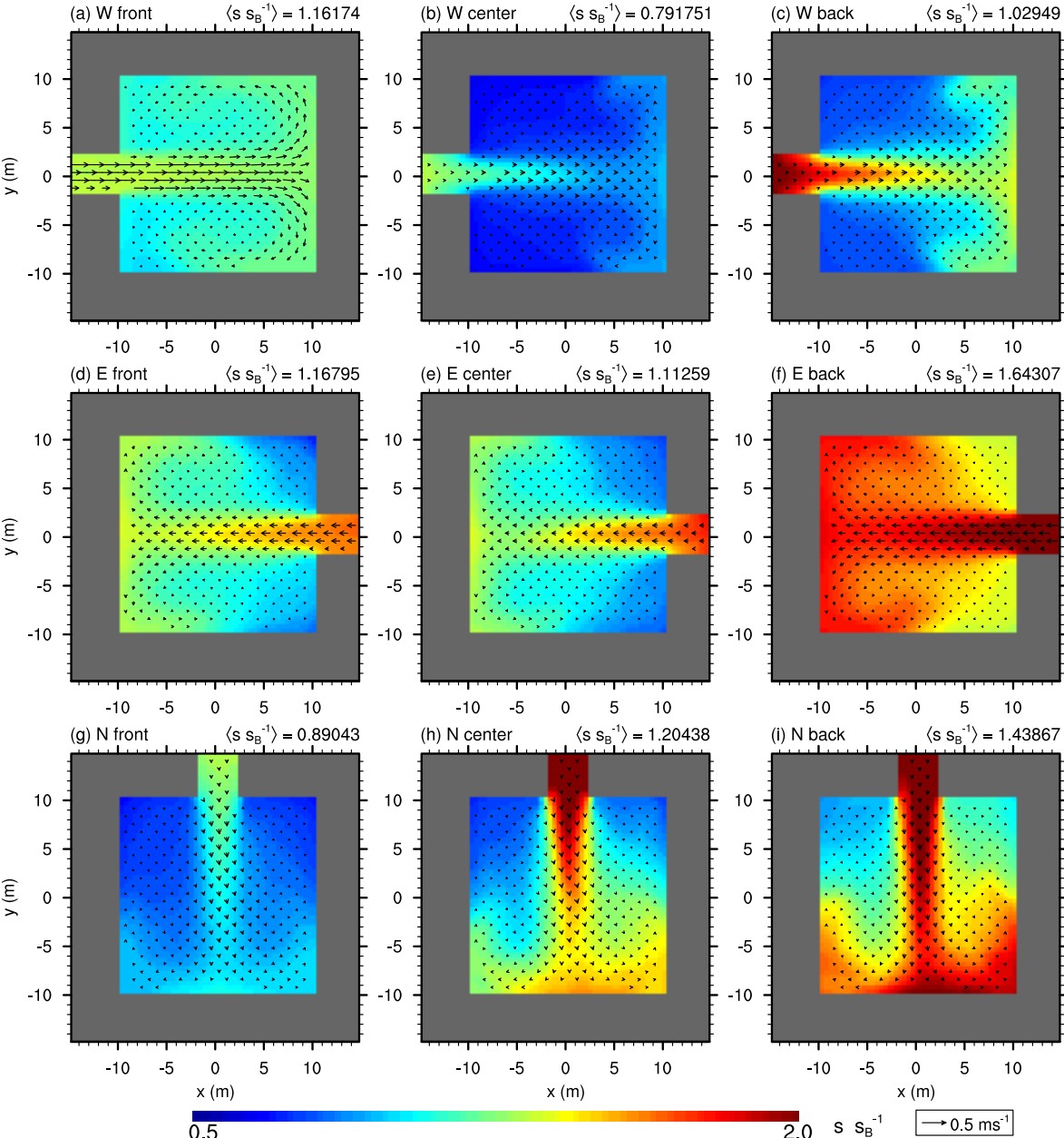

**Figure 8.** *Xy*-cross-section of the mean flow field (vector arrows) and mean scalar concentration (contours) within courtyards at $z = 1.8$ m for case AR3. Scalar concentration is normalized with the background concentration $s_B$ at $z = H$. Please note, for reasons of space, not all realizations are shown.

Courtyards with an eastward opening indicate an even stronger mean inflow (see Figure 8d–f), which comes together with high mean concentrations near the courtyard surface. The highest mean concentrations can be observed for courtyard "E back", which can be attributed to the near-surface re-circulation downstream of the building that entrains scalar through the opening into the courtyard.

The north/southward-opened courtyards also show a mean inflow into the courtyard, which is also accompanied by higher scalar concentrations. The scalar concentration within the courtyard increases from the front- towards the back-row courtyard (Figure 8g–i), which can be attributed to the increasing scalar concentration along the *x*-parallel street canyon, as indicated by Figure 6.

### 3.1.3. Case AR03

Figures 9 and 10 show vertical and horizontal cross-sections of the mean flow and scalar concentration for the wide courtyard setup AR03, respectively. As expected, the flow pattern and scalar distribution within the street canyons is comparable to case AR1. Within the closed courtyards, however, the re-circulation is shifted towards the windward building wall and covers only about two thirds of the cavity along the *x*-direction, while mean scalar concentrations are comparable to those in case AR1. Furthermore, in Figure 10a it strikes that near the leeward corners of the courtyard a pair of horizontally re-circulating eddies can be observed.

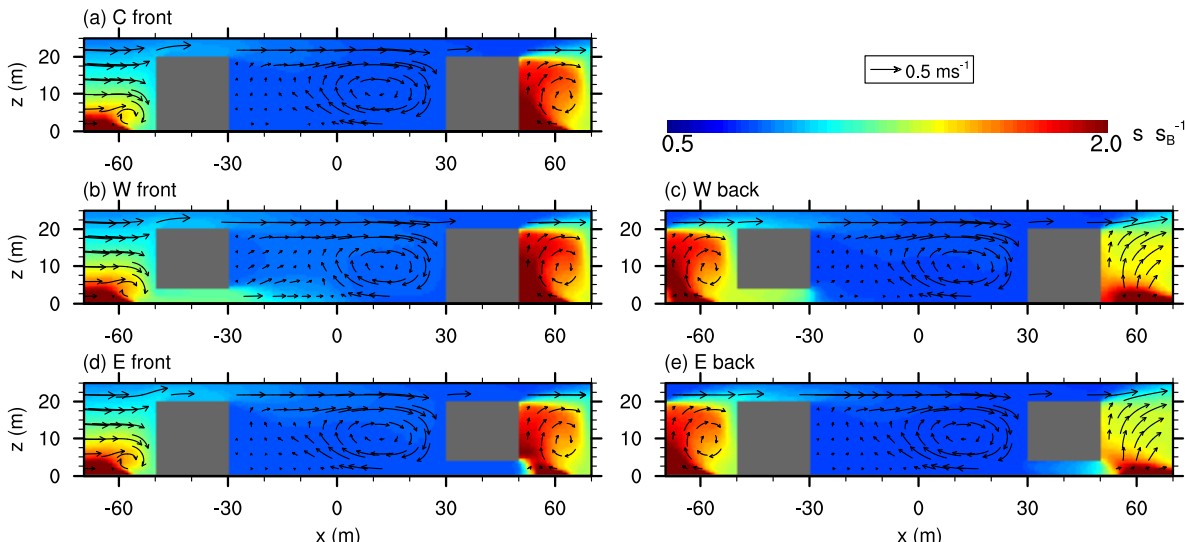

**Figure 9.** *Xz*-cross-section of the mean flow field (vector arrows) and scalar concentration (contours) at the courtyard center for case AR03. Please note, for reasons of space, not all realizations are shown.

Similar to the other aspect-ratio cases, "W front" shows a mean inflow through its opening, strengthening the pair of horizontal eddies near the surface and reaching into the courtyard about half its width (Figure 10b). The re-circulation (Figure 9b) within the courtyard, however, is less affected compared to case AR1 as it is already shifted towards the windward building wall. The mean scalar concentration shows higher values about half way into the courtyard near the surface, before it is transported upwards by the re-circulation and further mixed.

"W back" exhibits high scalar concentration along the tunnel-like opening. This actually contradicts with the re-circulation in the upwind street canyon that counteracts an inflow through the opening (similar to case AR1). However, the pair of horizontal eddies within the courtyard might promote the weak inflow of polluted air into the cavity (see Figure 10c).

Furthermore, it surprises that courtyards with an eastward opening do not show high concentrations, which is in contrast to case AR1 and AR3 (compare Figure 10d and, e.g., Figure 5g), even though the re-circulation pattern within the relevant street canyons are similar in shape and

strength compared to case AR1. Even a weak outflow through the eastward opening can be observed in Figure 10d (also present but weaker for "E back").

Results for courtyards with northward and southward openings (not shown) are similar to those of case AR1, except for the mean scalar concentration which is lower due to the larger courtyard volume compared to case AR1.

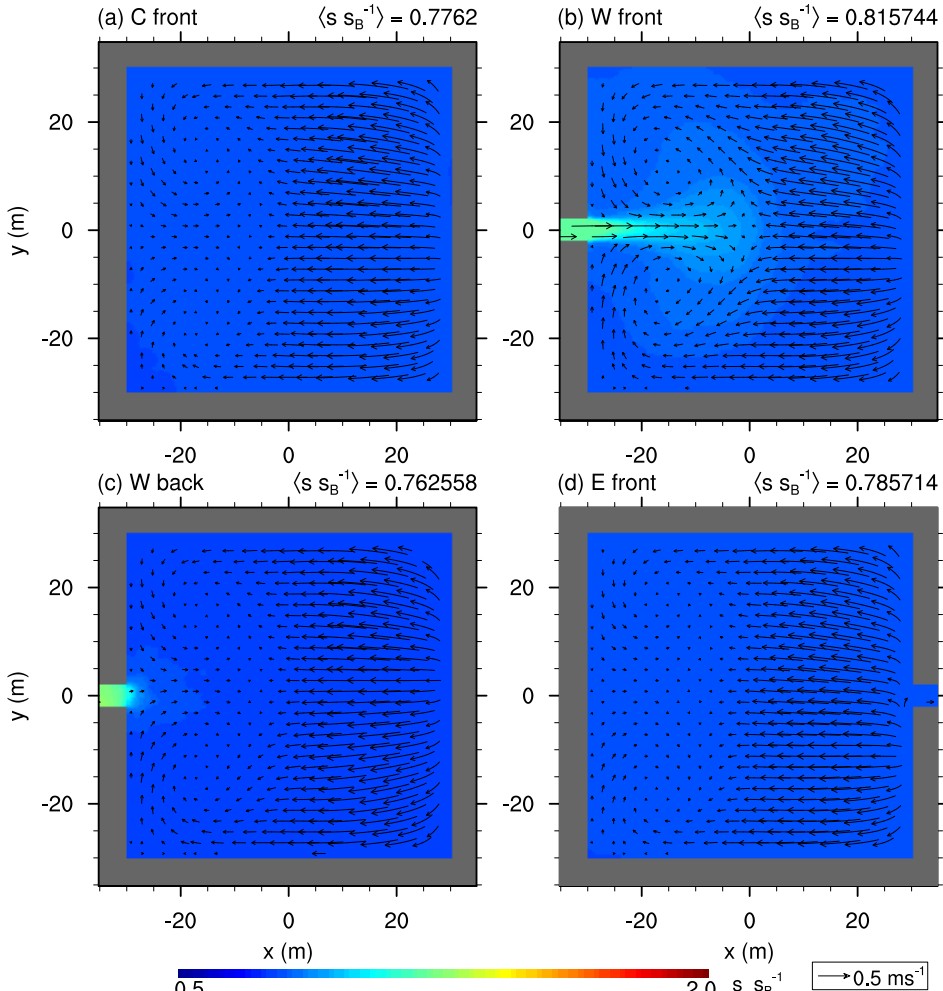

**Figure 10.** *Xy*-cross-section of the mean flow field (vector arrows) and scalar concentration (contours) within courtyards at $z = 1.8$ m for case AR03. Scalar concentration is normalized with the background concentration $s_B$ at $z = H$. Please note, for reasons of space, not all realizations are shown.

## 3.2. Quantification of Net Scalar Transport

In Section 3.1, we showed that the mean scalar concentration within the courtyards can be quite different among the different courtyard realizations, depending on the orientation of the opening, the incoming flow, as well as the strength and shape of the street-canyon circulation. The mean concentration, however, depends not only on the amount of entrained scalar into the courtyard but also on how long the scalar resides within the courtyard volume. Hence, to quantify how much scalar is transported through the lateral openings and thus estimate their significance on courtyard pollution, we calculated the net transport of scalar through the vertical and lateral openings as described in Section 2.2.

Figure 11 shows the time-averaged net transport of scalar into the courtyard volume by the lateral and the vertical opening for the different courtyard realizations. Positive values indicate increasing scalar concentrations within the courtyard and vice versa. As expected, the closed courtyards (black symbols) show slightly positive net transport through the top opening, which indicates slightly

increasing concentration within the courtyard cavity during the analysis period. For all ARs, the highest net transport by the lateral opening can be observed for the "W front" courtyards (blue circle, Figure 11), attributed to the non-blocked incoming flow. At this point, we want to emphasize that a high net transport into the courtyard does not necessarily correlate with high scalar concentrations. This becomes obvious for "W front" and "E back" in case AR3 (blue circle and green square, Figure 11a), where "W front" exhibits higher net transport than "E back" but shows lower mean concentration (see also Figure 8a,f). This is because the high net transport through the lateral opening for "W front" is relatively quickly compensated by the vertical transport across the top opening, while for "E back", the entrained scalar resides for a longer time within the courtyard leading to higher mean concentrations.

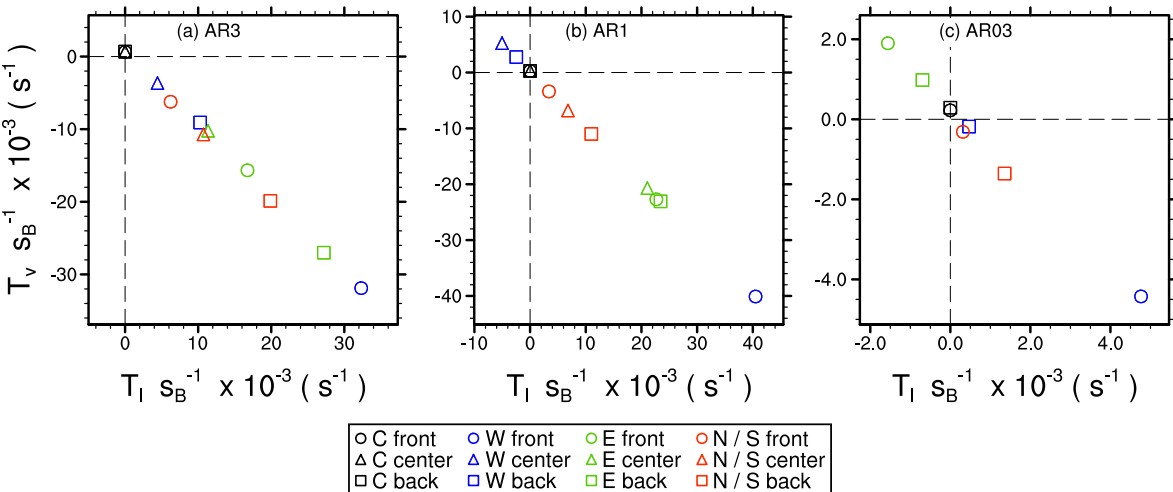

**Figure 11.** Net transport of scalar into the courtyard volume through the lateral (abscissa) and the top opening (ordinate), for (**a**) case AR = 3, (**b**) AR = 1, and (**c**) case AR = 0.3, averaged over 2 h of simulation time. Net transport is normalized with the background concentration. Positive (negative) values of net transport indicate increasing (decreasing) scalar concentration within the courtyard cavity. The dashed horizontal and vertical lines indicate zero values.

For the "W center" and "W back" courtyards (blue triangle and square, Figure 11) the situation is different among the different ARs. In case AR3 and AR03 (Figure 11a,c), scalar is entrained through the lateral opening and detrained through the top opening, while in case AR1 (Figure 11b), these courtyards show a net detrainment of scalar through the lateral opening and a net entrainment of scalar through the top opening, according to the mean concentration level shown in Figure 4.

For cases AR3 and AR1, the courtyards with eastward openings (green symbols, Figure 11a,b) show positive net transport through the lateral opening. In contrast, for case AR03 (Figure 11c), the eastward-opened courtyards show a negative net transport of scalar through the lateral opening. As we already mentioned in Section 3.1, this is in contradiction to the shape of the circulation patterns within the courtyard and the adjacent street canyon, which both points towards the courtyard center and thus should promote the transport of scalar-rich air into the courtyard (see Figure 9d,e). This gives rise to the question of how important theses circulations are for the transport of scalar through the lateral opening. Here, we must note that investigating this question is beyond the scope of this study. Hence, we postpone a more detailed analysis of this contradiction, as well as a quantitative analysis of the relevant transport mechanisms (re-circulation or random turbulent mixing) responsible for the lateral mixing of scalar into the courtyard to a follow-up study.

The north/southward-opened courtyards (red symbols) indicate a net transport through the lateral opening for all ARs, which is in accordance to the high mean concentrations within the tunnel-like opening (see, e.g., Figure 8). It strikes that the net transport through the lateral opening is lowest for the front-row courtyard and highest for the back-row courtyard. This is related to

the increasing scalar concentration along the *x*-parallel street canyon, with the largest gradient in concentration between street canyon and courtyard in the back row.

### 3.3. High Scalar Concentration Events

The previous analysis revealed that mean scalar concentrations within the courtyard cavities are lower compared to the adjacent street canyons. To estimate the impact on human health, however, the exposure to high concentrations and their relative occurrence are also relevant. To investigate whether lateral openings promote the occurrence of high concentrations within courtyards, we calculated PDFs from the sampled scalar concentrations at the courtyard center at a height of 1.8 m (representative for human exposure). Figure 12 shows the PDFs for the different ARs and courtyard setups. Among the different ARs, the closed courtyards (black curves) show similar PDFs with quite narrow Gaussian-shaped distributions and median values below the background concentration, indicating that almost no high concentrations are mixed from the top into the courtyard. For case AR03 (Figure 12c), almost all courtyard geometries show PDFs which are quite similar to the closed-courtyard PDFs. Only the "W front" courtyard (solid blue curve) shows a slightly skewed PDF where scalar concentrations reach values up to 1.5 times the background concentration, which is due to the direct inflow of scalar-rich air from the street canyon through the lateral opening. Similarly skewed PDFs can be observed for the "W front" courtyards in case AR3 and AR1 (Figure 12a,b).

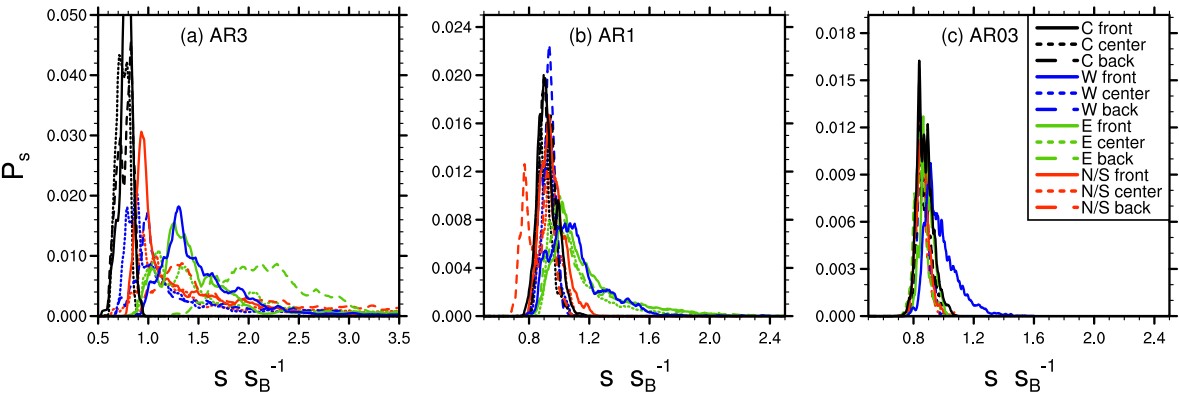

**Figure 12.** Probability density function of scalar concentration at courtyard center at $z = 1.8$ m for (**a**) case AR = 3, (**b**) AR = 1, and (**c**) case AR = 0.3. The scalar concentration is normalized with the background concentration $s_B$, which is the domain-averaged concentration at $z = H$.

In case AR1 (Figure 12b), also the eastward-opened courtyards (green curves) exhibit PDFs that are skewed towards higher concentrations, which is in accordance with the increased lateral mixing of scalar into the courtyard via the lateral opening (see Figures 5 and 11). In the high aspect ratio case AR3 (Figure 12a), all open-courtyard setups show skewed PDFs that tend towards higher concentrations, with maximum values multiple times the background concentration. Especially "E back" (green dashed curve) as well as the north/southward-opened courtyards (red curves) show a significant number of high-concentration events, according to the high lateral net transport of scalar into the courtyard (see Figure 11a).

### 3.4. Residence Time of Pollutants

To estimate for how long scalar resides within the courtyards, we applied a Lagrangian particle model embedded into the LES and evaluated of how long particles reside within the courtyard cavity. Figure 13 shows the PDFs of particle residence times for the different courtyard realizations. In case AR03 (Figure 13c), as expected, no significant differences between the different courtyard setups can be observed, indicating that the overall impact of lateral openings onto courtyard ventilation is negligible in this case as all courtyards are already well ventilated by the large top opening. The PDFs

are skewed towards larger residence times, with a small number of particles that reside up to 2000 s within the courtyard; we could trace this back to particles that were trapped mainly near the corners of the courtyard within the pair of horizontal eddies which are most pronounced for "W front" (see Figure 10). Most particles, however, reside only for a few hundreds of seconds within the courtyard, with peak positions at about 100 s (see Figure 13c).

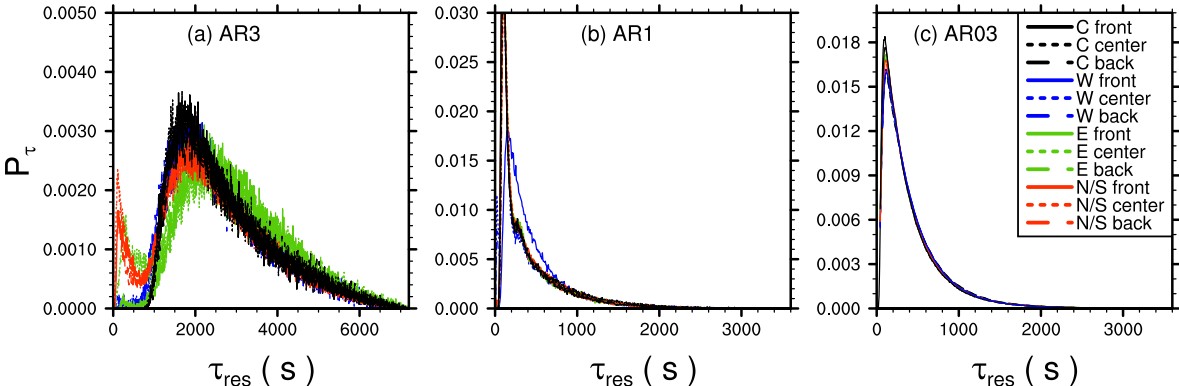

**Figure 13.** Probability density functions of particle residence times within the courtyard volume, for (**a**) case AR3, (**b**) AR1, and (**c**) case AR03.

Similar PDFs can be observed for case AR1 (Figure 13b), with no significant differences among the courtyard setups, except for the "W front" courtyard (solid blue curve) with slightly higher residence times. There, the re-circulation within the courtyard cavity is disturbed by the lateral inflow and does not reach down to the surface so that vertical particle transport in the lower part of the cavity is less effective. Taking particle residence times as a direct measure of ventilation, this means that "W front" openings even slightly decrease courtyard ventilation in case AR1, which agrees with the findings of Hall et al. [2] who also observed less ventilated courtyards if the flow can directly enter the courtyard through the opening.

For the high aspect ratio case AR3 (Figure 13a), the peak positions are shifted towards longer times of about 1800 s to 2300 s, meaning that particles reside for a significantly longer time period within the courtyard compared to the lower aspect-ratio cases.

This is not surprising for two reasons: first, the vertical distance between the particle source and the particle sink at the top opening is larger compared to AR1 and AR03; and second, the flow near the surface is detached from the upper part within the courtyard as shown in Figure 7, which reduces turbulent mixing within the lower part of the courtyard. For AR3 (Figure 13a), it strikes that "E center" (dotted green curve) as well as the north/southward-opened courtyards (red curves), exhibit a secondary peak at about 100 s, with a significant number of particles reside only for a short period of time within the courtyard cavity, while the bulk of particles reside for longer period of time. This secondary peak indicates intermittent mixing events where particles are quickly mixed out of the courtyard at some point in time; in fact, we could observe intermittent events in the scalar timeseries sampled at the courtyard center for these courtyards (not shown). At this point, however, we note that the physical mechanisms that promote this behavior for these building setups are not clear, drawing the need to investigate this in more detail in follow-up studies.

Furthermore, we note that the larger residence times observed in case AR3 come together with the increased number of high-concentration events as discussed in Section 3.3. This, in turn, suggests that scalar that is once mixed into the courtyards remains there for a longer time, increasing the human exposure to pollutants in such courtyard geometries.

## 4. Conclusions

We performed a set of LESs for idealized building-courtyard setups where we altered the height-to-width ratio of the courtyard cavities. Based on this, we studied the effect of tunnel-like

lateral openings onto courtyard ventilation and pollution. We considered different orientations of lateral openings with respect to the mean flow direction, as well as different arrangements of the building blocks, with non-blocked and blocked oncoming flow in the front- and center/back-building row, respectively.

To estimate courtyard ventilation, we calculated residence times using a Lagrangian particle model embedded into the LES. Courtyards with an AR of one (courtyard width equals courtyard depth) and wide courtyards show similar residence times with mean values of a few hundreds of seconds, indicating that both geometries are similarly ventilated. In contrast, deep courtyard geometries show significantly larger residence times with mean values of more than half an hour. We showed that lateral openings can affect courtyard ventilation in different ways, ranging from increasing to decreasing the ventilation. Wide courtyards show almost no impact of lateral openings on ventilation as these are already well ventilated via the top opening. Also, courtyards with an AR of one show no significant effect of lateral openings onto the ventilation, except for windward lateral openings in the front row. There, courtyard ventilation is slightly decreased since the re-circulation within the courtyard cavity is partly disturbed by the direct lateral inflow, so that the vertical exchange within the lower half of the cavity is less effective, which is in accordance to the results shown by Hall et al. [2].

In terms of ventilation, the deep courtyards indicate a more complex behavior. Courtyards with windward openings as well as leeward openings in the front and center row show similar residence times than closed courtyards. This is different for lateral openings with orientation perpendicular to the mean flow as well as leeward-orientated openings in the back row. There, the residence times indicate the occurrence of two alternating regimes, one similar to that in the closed-courtyard case with large residence times indicating poorly ventilated courtyard cavities, and a second one with low residence times indicating well-ventilated cavities. However, within this study we did not analyze this in more detail, nor did we figure out what are the responsible mechanism that promote such intermittent mixing events. This must be part of future research.

To study the effect of lateral openings onto courtyard pollution, we emitted a passive scalar along the center of the street canyons, emulating pollution by traffic. For the deep courtyards, on average, scalar is entrained through the lateral opening and detrained through the top opening for all considered courtyard realizations. The largest entrainment can be observed through the windward openings in the front row, where the flow can enter the courtyard unhinderedly. Courtyards with leeward openings in the back row show also large entrainment of scalar through the lateral opening, as the re-circulation downstream of the building transports high scalar concentrations towards the opening. Likewise, deep courtyards with openings orientated perpendicularly to the mean flow are strongly polluted as a mean inflow develops from the street canyon into the courtyard.

The large entrainment also causes deep courtyards with lateral openings to show a significant increase of mean scalar concentration as well as an increased number of high-concentration events compared to closed courtyards. This, in turn, reveals the negative impact of lateral openings on the air quality in deep courtyards. In addition, this comes together with large particle residence times in deep courtyards, indicating that once high scalar concentration is mixed into the courtyard cavity, it stays therein for a while. However, to estimate the impact on human health issues, future studies need to draw a clear connection between high scalar concentrations and large particle residence times, where this study was not designed for. For example, suppose high concentrations are recurrently entrained into the courtyard but mixed out relatively quickly, while low concentrations remain in the courtyard cavity for a longer time. Now, suppose a situation where high concentrations are only entrained into the courtyard occasionally but mixed out very slowly. Both situations could lead to similar particle residence times, high-concentration events, and mean concentrations but may have a significantly different impact on human health, e.g., [4,5,15].

For courtyards with an AR of one, scalar is entrained through the lateral opening and detrained through the top opening for most of the courtyard realizations. Largest entrainment could be observed through the windward openings in the front row and through the leeward openings, where also the

highest mean concentrations as well as the highest number of high-concentration events could be observed. For windward lateral openings in the center and back row, however, the situation becomes different and, on average, scalar is detrained through the lateral opening and entrained through the top opening. This is mainly attributed to the re-circulation within the crosswind-aligned street canyon and the courtyard circulation which cause a mean outflow through the opening.

For wide courtyards, again, the largest net entrainment through the lateral opening can be observed for windward openings in the front row, while detrainment through the lateral opening can be observed for the leeward openings.

To summarize, the effects of lateral openings on courtyard pollution and ventilation are diverse. In general, it can be said that the influence of lateral openings weakens for decreasing AR (courtyards become wider). For courtyards with low AR, the ventilation and pollutant de- and entrainment through the top dominates, so that laterally opened courtyards exhibit very similar flow patterns and scalar distributions as closed courtyards. In contrast, for high ARs, lateral openings strongly alter the ventilation and air quality within courtyards, as the bottom part of the courtyard cavity is poorly ventilated from the top opening.

We note that our idealized study only covers a small part of possible courtyard configurations and air pollution scenarios. For example, different opening sizes or opening locations that are not centered may also affect the entrainment of scalar into the courtyards, as well as the flow within the courtyard and thus its ventilation. Hall et al. [2] had shown that also solid obstacles at the courtyard surface decrease the ventilation and increase concentrations near the surface. We expect the same for plant canopy located within courtyards, which would perturb the courtyard circulation and suppresses turbulent mixing. This case, even wide courtyard geometries may become poorly ventilated by the top opening, so that the impact of lateral openings on air pollution gains further relevance.

In this study, we considered only scalar sources on the streets. However, inner-courtyard sources (courtyards are also often used as car parks) or domestic fuel may also be important with respect to the air quality within courtyards. Hence, lateral openings may possibly become even important for scalar removal from courtyards.

Also, buildings and roof shapes, e.g., [13], as well as variable building heights, e.g., [12,14] may alter courtyard ventilation, to name only a few of possible parameters.

Another aspect we neglected in our study are buoyancy effects. Thermodynamic effects can significantly alter ventilation patterns in street canyons [20] or even in entire cities [38]. We expect this will account for courtyards as well. Especially in warmer climates, where courtyards are often used to control indoor ventilation and cooling, different studies, e.g., [39] already revealed that courtyards of different depth can increase or decreased thermal ventilation.

**Supplementary Materials:** The complete list of parameters for all presented simulations as well as the modified code parts used in addition to the standard code base of PALM are available online at http://www.mdpi.com/2073-4433/10/2/63/s1.

**Author Contributions:** Both authors contributed equally to the study including conceptualization, methodology, software, formal analysis, visualization and writing. M.S. administrated the project and T.G. conducted the simulations.

**Funding:** This research was part of the MOSAIK project. MOSAIK is funded by the German Federal Ministry of Education and Research (BMBF) under grant 01LP1601A within the framework of Research for Sustainable Development (FONA; www.fona.de). The publication of this article was funded by the Open Access fund of Leibniz University Hannover.

**Acknowledgments:** The authors thank the two reviewers for their critical and valuable comments which helped to improve the manuscript. All simulations were carried out on the computer clusters of the North-German Supercomputing Alliance (HLRN). NCL (The NCAR Command Language, Version 6.3.0, 2015 (Software), Boulder, Colorado: UCAR/NCAR/CISL/VETS, http://dx.doi.org/10.5065/D6WD3XH5) was used for data analysis and visualization. The PALM code can be accessed under www.palm-model.org.

**Conflicts of Interest:** The authors declare no conflict of interest. The funders had no role in the design of the study; in the collection, analyses, or interpretation of data; in the writing of the manuscript, or in the decision to publish the results.

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
