# Peer review of "On the Effects of Lateral Openings on Courtyard Ventilation and Pollution—A Large-Eddy Simulation Study"

_atmosphere, doi:10.3390/atmos10020063_

Round 1
Reviewer 1 Report
Paper reads better now the reviewers comments have been added. Work is stronger, with plots easier to understand and the phrasing more formal. The added references and extra material around the model further strengthen the work.
I am happy for the journal to accept this article in its current form and look forward to future work from the authors.
Author Response
We would like to thank the reviewer for again reviewing our revised manuscript.
Yours
sincerely,
Tobias
Gronemeier
Reviewer 2 Report
I would like to thank the author for implementing my comments. I still have one comment and one concern before the paper can be accepted for publication:
1) As I mentioned in my previous review (comment#2), it is important to provide more information about the computational grid. A figure should be also added to better show the details of the grid that is of utmost importance for LES simulations.
2) Line 146: The authors should double-check the number of cells used in the simulations. I think the authors are very well aware that it would be almost impossible to perform LES simulations on such a high-resolution computational grid with about 720 million cells, though the simulations have been performed on a supercomputer
Author Response
@page { size: 210.01mm 297mm; margin: 20mm } p { margin-bottom: 2.47mm; line-height: 115%; background: transparent } a:link { color: #000080; so-language: zxx; text-decoration: underline }Dear reviewer,
We would like to thank you again for reviewing our revised manuscript.
You gave two comments to which we reply in the attached PDF file. For better reference, comments are repeated and written in italic while responses are written in regular.
Yours
sincerely,
Tobias Gronemeier

This manuscript is a resubmission of an earlier submission. The following is a list of the peer review reports and author responses from that submission.
Round 1
Reviewer 1 Report
This is a very interesting paper on a topic of very large importance to the urban physics and building physics community. As such, it will be a very welcome contribution to the journal. However, some mandatory revisions should be implemented before it can be accepted:
1) Concerning the building models (Fig.1), I think it would be better to add a 3-dimensional (3D) figure showing the generic urban model. Although the authors presented a 2-D figure along with the long text about it to the paper, it would be a bit difficult to follow and easily understand which is which.
2) The section on the computational grid is incomplete. Further information about the total number of cells, the average and maximum y* values, type of the grid used, etc., should be added. In addition, the authors should show a figure showing the computational grid.
3) Line 123: it is mentioned a grid-sensitivity analysis is performed, but the results are not shown here. I think the quality of the computational grid is essential for LES simulations, and I highly suggest that the authors add this to the paper.
4) Lines 123-125: It is mentioned, “A sensitivity study (not shown) indicated this grid spacing to be sufficient as turbulence statistics within the courtyards, the street canyons as well as within the tunnel-like openings did not change for further decreasing grid spacing”. As the authors are very well aware, with LES and implicit filtering, the model depends inherently on the grid size. When refining the grid, the model contribution is also changing and consequently, a grid-independent solution cannot be found. I highly recommend that the author take a look at the following position papers on LES simulations:
- Freitag M, Klein M. An improved method to assess the quality of large eddy simulations in the context of implicit filtering. J Turbul 7 (2006) 1–11.
- Pope SB. Ten questions concerning the large-eddy simulation of turbulent flows, New J. Phys. 6 (2004) 35.
- Iousef S, et al. On the use of non-conformal grids for LES of flow field and convective heat transfer for a wall-mounted cube. Building and Environment 119 (2017) 44-61.
5) Line 144-148: to investigate pollutant dispersion in the domain, line sources of passive scalar are considered within the street canyons. I am a bit concerned about how realistic it would be to model identical surface fluxes on all streets. As far as I know, recently a new concept so-called “air delay” has been developed that could be more relevant for such studies. This should be better clarified in the paper. The authors should also justify in the paper what would be the advantages and disadvantages of this modeling approach compared to other approaches, in particular, the “air delay” that deals with the freshness of the air. The latter could be relevant especially when the focus is on pollutant residence time as it is correctly mentioned in Sec. 2.3.
6) The validation study does not contain pollutant concentration, but only wind flow. Since the accuracy and reliability of CFD results are always of concern and detailed validation studies are imperative, the authors should at least refer to similar studies in which a detailed validation study has been performed. This should also clearly be stated as one of the limitations of this study.
7) Although the authors considered different cases, and analyzed the results in detail for all cases, I think there is a serious lack of effort in generalizing the work for the scientific community. This should be added to the paper.
Other comment:
- Line 216: “theses” should be “these”.
Author Response
@page { size: 210.01mm 297mm; margin: 20mm } p { margin-bottom: 2.47mm; line-height: 115%; background: transparent }Dear reviewer,
We would like to thank you for carefully reviewing our manuscript and providing us with useful comments and valuable suggestions that substantially helped to improve the quality of the manuscript.
A detailed list of all given comments and questions (written in italics) and our response to those comments (written in regular) is given in the uploaded PDF file.
Yours sincerely,
Tobias Gronemeier

Reviewer 2 Report
Overall
An interesting concept that has clearly had a considerable amount of time spent on it. The paper highlights the considerable risk of trapped pollutants within courtyards and how the orientation of the courtyard influences flow and pollutant residence time. The use of an external source makes it unique and of interest to urban planners with the findings possibly being applicable to other sectors with regards to building orientation and entrainment into buildings. Plot quality and research quality is good, with the paper being easy to read. The paper highlights the use of PALM and validates based on wind tunnel experiments to ensure scientific robustness. The paper also is clear on what further research can be undertaken and of its own limitations.
I have provided specific comments and requests for information that I feel will strengthen the paper and make it more robust. I hope they help you and are clear.
----------------------------------
Overall points:
Be consistent on whether you’re using Sect. or Section in the text.
Consider changing plot colour schemes to remove either red or green to improve accessibility to those with vision impairments. Also- none of your plots feature the dark blue (near 0 values), consider limiting the colourbar to allow for more detail to be visible in your plots.
Some of the language used is quite casual- this needs to be tightened up throughout. Some sentences are quite wordy for the message they’re conveying, so please go through and tighten up.
I’m not keen on the use of subsubsections when the sections themselves contain no text- this is personal preference though, so do what works for you.
You have some excellent plots so do reference them more throughout the text to back up your statements. You do this in some places but not others. Also state earlier on that not all variations (NESWC etc) are shown and give a reason why (eg nothing of interest, similar to another case etc). Don’t talk about plots that you don’t show- as if they’re interesting, you should be showing them.
With regards to LES, please put in as much information as possible about the simulations etc, so that someone with no prior knowledge of PALM could accurately recreate the experiment. This can be in the form of extra references, or tables of key values (my specialism is not LES).
----------------------------------------------------------------------------------
Introduction
Overall:
A statement about how rare work into flow and concentrations within courtyards is would help strengthen your reasons for research. A few more references sprinkled in would also strengthen it.
I would consider rearranging some of the paragraphs, and discuss the human health impacts first (line 59, move up to line 22), then go into detail about the previous research undertaken.
Specifics:
First lines of the introduction (line 18) and the abstract are exactly the same. Please change the wording of one.
Outer sources- external sources may be a better way of phrasing it
‘Courtyards, however, are often poorly-ventilated spaces [1], where contaminants from outer sources that are once mixed into, or local sources such as exhausts from domestic fuel or cars, can pose a serious threat to human health.’
Just check the grammar of this sentence so that its clear. Maybe consider splitting the sentence into two- first sentence discussing the fact that courtyards are poorly ventilated and pollutants become trapped and then a second sentence saying the sources of these pollutants (with a reference).
The review of previous research reads a bit like ‘these people did this and then that’ etc. Weaving all the research together and removing some of the experimental details to identify the knowledge gaps would help.
Line 25- Several studies ‘have’ already… I’d like to see a few more references here
Also, consider bullet pointing your aims so that they are extra clear to the reader (line 67). I’m not keen on them being presented as questions as I feel they are stronger as statements, but that is just personal preference
Line 76- ‘The paper is organized as follows:’ can be deleted.
Figure 1- needs to be below the Section 2 header, this is likely LaTeX formatting gone awry so check this. Figure 1 caption- Different shades of grey indicate building patches with ‘a’ different courtyard configuration, labelled according to their lateral opening orientation. The inclusion of the red colour bar is confusing and not explained in the caption (as all lines appear the same colour red in my copy), so consider removing/double check.
--------------------------------------------
Methods
I want enough detail in this section to be able to repeat your experiment from scratch, or a reference to a more detailed source/ data storage. Explanation of the choice of PALM would also be helpful to the reader. Some examples of other work using PALM would also help as would pros and cons etc (these could be moved from your validation section)
I understand the reasoning for the naming convention of AR1, AR3 and AR03, but it is confusing at times so ensure that it is clear throughout the paper and reference table 1 and figure 1 as often as possible.
I’d like to see an explanation for why these aspect ratios were chosen within the paper much like your explanation for the configurations. Also an explanation for the chosen opening size and why the certain locations.
Consider including the domain sizes in table 1 for a more complete overview of the cases
A table of the key facts about the simulation (spin up time, speed, shear driven only etc) would also help the reader and would reduce the word count of this section.
A statement describing how the layout may not be realistic and how the idealised layout you have will further the research into the subject area is required- this could come from your conclusions section
Line 102- it gets a bit confusing with the brackets for AR03 for the reader- just write two sentences one for AR1 and AR3 and one for AR03. This then makes the following statements in the paper easier to understand.
Line 118- delete ‘of course’
Line 130- give an error/ stdev on this figure for wind speed.
I would maybe have a split subsection for flow and then one for concentration
Line 151- put analysis period time in brackets for extra clarity.
Line 160- reference figure 1 again at the end of the line
Line 168 & 170- terms instead of term
Equation 1- please define all symbols to make it easier for readers unfamiliar with the topic
Line 187/188 ‘In other words, once a pollutant enters the courtyard, its impact 188 on human health depends on the time required to exit the courtyard again.’ Can probably be deleted
Line 188- delete ‘these’
Line 199- a new paragraph somewhere in this area would help the reader
Line 209- is removing the particle from the simulation realistic? In a real urban area it may get reentrained. A comment about why this method was chosen would help (or moving the comment from the end of the section up here)
Line 223- I don’t like the phrase ‘for the sake of correctness’, please make more scientific.
-----------------------------------------------
Results
Figure 2- weird formatting on the legend for Hall. The ll appears as a solid block in my PDF. The hashing makes the black curve difficult to see- maybe change to a lighter grey/ up the transparency/change the plot order? Explain the hashings in the caption.
‘Profiles are time-averaged over 3 h.’ how long was the validation run for and is this true of all the data on the graph? Please add in additional statements to clarify this- people need to have enough information to repeat the experiments.
A statement on the errors/stdev of the measurements is also required in the text, even if to state that errors/stdev are not given in the comparative studies.
It might make the paper clearer if you move the validation into the methods section and begin the results section with subsections for the different cases.
Line 252- consider referencing figure 1 and table 1 again
Line 253- give examples of the flow behaviour you’re discussing in this section using the figures as this will strengthen your results e.g.-
Within the courtyard cavities, well-defined re-circulations are present which extend throughout the entire cavity (e.g. Figure 3x).
You do do this later on and that paragraph is much stronger as a result.
Figure 3- Why are north and south not included? Give a statement to make it clear to the reader what they’re looking at
Figure 4- text of what plot is what is quite small consider doing text at the top of the figure and to the left or right of the figure to create a grid- so Front, centre and Back on the top, then the locations on the side (C,W, E, N). Also consider referencing figure one in the caption. As the x and y axis are all the same, consider just labelling the far left plots with the x labels and the bottom plots with the y labels to give more space in the figure. Why are the South courtyards not included? Again, give the reason.
Line 273- You say ‘see Figure 3d and 3d’- check referencing here.
Line 287- If the opening is located at either the northern or southern wall, the x-parallel street-canyon flow 288 slightly pushes polluted air into the courtyard (cf. Fig. 4 j to l)-> you need a statement either explaining that north and south look the same, or check that you don’t need to include south plots in your figure
Line 296- ref to table 1 at the end of the sentence
Line 299- reference one of the plots that shows this
Line 304- what’s the reason for not showing this plot?
Line 316- A new paragraph here will help the reader
Line 322-324- give examples of the plots where you see this for both cases
Figure 7- bit late to be saying this statement- say it at the start of the results and consider including extras in supplementary material or making them available in the doi. Also needs to come after the AR03 section starts though this is probably a LaTeX figure placing issue.
Line 342- space between Figure and 7 required.
Figure 8- colour bar does not help this plot- limit the colourbar range to get more detail in the plots
Line 376- check spacing between figure and 9
Figure 9- legend is too small, consider pulling it out of the plot and putting it to the side. Bigger font for axes. Black icons need to be more distinct
Lines 380 onwards- reference more to figure 9- say W (blue triangles) to make it easier for the reader
Line 383-384 ‘, as mentioned at the beginning 384 of this paragraph.’ – not needed
Line 388- (please see discussion in Sect. 3.4). Get rid of this as you’re referencing forward.
Figure 10- bigger axis labels will help with clarity. Needs to be in section 3.4
Line 413 onwards- again reference more to your figures when you talk about your cases- also true for discussion of figure 11 in the next section
----------------------------------------------------------------
Summary and Conclusions
Just call it conclusions.
I think this section can be a bit more consise and link back to your key aims as most of this is said previously in the discussion of the results. Use the aims to structure your conclusions, summary and future work. Good discussion of limitations but this may be better earlier in the paper in a subsection within the methods.
Author Response
Dear reviewer,
@page { size: 210.01mm 297mm; margin: 20mm } p { margin-bottom: 2.47mm; line-height: 115%; background: transparent } a:link { color: #000080; so-language: zxx; text-decoration: underline }
We would like to thank you for carefully reviewing our manuscript and providing us with very useful comments and valuable suggestions that substantially helped to improve the quality of the manuscript.
A list of all given comments and questions (written in italics) and our response to those comments (written in regular) is given in the uploaded PDF file.
Yours sincerely,
Tobias Gronemeier
